# Adenovirus Transcriptome in Human Cells Infected with  ChAdOx1-Vectored Candidate HIV-1 Vaccine Is Dominated by High Levels of Correctly Spliced HIVconsv1&62 Transgene RNA

**DOI:** 10.3390/vaccines11071187

**Published:** 2023-07-01

**Authors:** David A. Matthews, Rachel Milligan, Edmund G. Wee, Tomáš Hanke

**Affiliations:** 1School of Cellular and Molecular Medicine, Faculty of Life Sciences, University of Bristol, Bristol BS8 1TD, UK; rachelmilligan54@gmail.com; 2The Jenner Institute, Nuffield Department of Medicine, University of Oxford, Oxford OX3 7DQ, UK; edmund.wee@ndm.ox.ac.uk; 3Joint Research Center for Human Retrovirus Infection, Kumamoto University, Kumamoto 860-0811, Japan

**Keywords:** adenovirus transcriptome, HIVconsvX, HIVconsv, HIV vaccines, RNA splicing, ChAdOx1, ChAdOx1.tHIVconsv1, ChAdOx1.HIVconsv62, C1, C62

## Abstract

We develop candidate HIV-1 vaccines, of which two components, ChAdOx1.tHIVconsv1 (C1) and ChAdOx1.HIVconsv62 (C62), are delivered by the simian adenovirus-derived vaccine vector ChAdOx1. Aberrant adenovirus RNA splicing involving transgene(s) coding for the SARS-CoV-2 spike was suggested as an aetiology of rare adverse events temporarily associated with the initial deployment of adenovirus-vectored vaccines during the COVID-19 pandemic. Here, to eliminate this theoretically plausible splicing phenomenon from the list of possible pathomechanisms for our HIV-1 vaccine candidates, we directly sequenced mRNAs in C1- and C62-infected nonpermissive MRC-5 and A549 and permissive HEK293 human cell lines. Our two main observations in nonpermissive human cells, which are most similar to those which become infected after the intramuscular administration of vaccines into human volunteers, were that (i) the dominant adenovirus vector-derived mRNAs were the expected transcripts coding for the HIVconsvX immunogens and (ii) atypical splicing events within the synthetic open reading frame of the two transgenes are rare. We conclude that inadvertent RNA splicing is not a safety concern for the two tested candidate HIV-1 vaccines.

## 1. Introduction

The overall safety profile of licensed vaccines is typically thoroughly established, and their risk–benefit ratio is well in favor of the latter. As of January 2023, the administration of over 13.2 billion of vaccine doses against COVID-19 inevitably associated vaccines with undesirable rare adverse events (AEs) [1,2,3,4,5]. The background incidence rates of various conditions in the population and the large numbers of individuals being vaccinated placed some of these AEs in the post-vaccination time window by chance. However, determining causal links is inherently difficult. On an individual case report level, the close temporal relationship between vaccination and the presenting symptoms, biological plausibility and extensive diagnostic efforts aiming to exclude other causes often fulfil the World Health Organization’s criteria for causality and lead to the assessment of these AEs as likely to be due to the vaccination [6]. On a population level, establishing causal links requires large epidemiological studies [7]. Thus, a claim for a causal association between the treatment and AE is always strongly supported by a pathomechanistic explanation for the reported symptoms. The absence of such an explanation diminishes the tested possibilities and strengthens the vaccine safety profile.

Adenoviruses are genetically stable viruses with a linear double-stranded DNA genome, which have been explored as vaccine vectors for several decades [8,9]. ChAdOx1 is a purpose-engineered vaccine vector derived from simian adenovirus 23 (Y25) of Mastadenovirus group E [10,11]. Deletion in early region E1 of the adenovirus genome rendered ChAdOx1 growth-deficient and further deletion of the nonessential E3 locus enlarged the “cargo” space for an insert coding for the vaccine immunogen to approximately 7 kilobase pairs [12]. Growth, including the Good Manufacturing Practice preparation of ChAdOx1-vectored vaccines for use in humans, is only possible in cell lines providing the essential E1 transcription factor function in trans [13,14]. In contrast, the ChAdOx1 vector cannot replicate and produce infectious progeny in the infected cells of vaccine recipients. Thus, the main function of ChAdOx1 is to efficiently deliver the pathogen-derived transgene into the host cells for the production of the protein immunogen. Such one-cycle vaccines mimic natural virus infection and efficiently induce both pathogen-specific CD8 T cells and antibodies [9,11,15,16,17,18,19,20,21,22].

RNA splicing was first discovered during studies of human adenoviruses, which are known for their complex splicing patterns of transcribed virus-derived RNAs [23,24,25,26,27,28,29,30,31]. We recently published an analysis of the human adenovirus type 5 (HAdV-5) transcriptomic repertoire showing that a super complex array of splicing and polyadenylation events is possible in adenovirus-infected cells [32]. This observation has been confirmed [33] and expanded on by others including showing how this kind of analysis reveals previously unappreciated fusion proteins [34]. Indeed, this complexity of splicing has also been seen in fowl adenoviruses [35]. It is therefore not surprising that alternative splicing through splicing signals inadvertently generated during the design of the immunogen-coding inserts was suggested as a safety risk [36]. These novel splice RNA products may be the cause of the overproduction of unusual splice variants or chimeric protein(s), which theoretically may lead to presenting AEs [37,38]. To better understand this risk, we published an analysis of the widely used SARS-CoV-2 vaccine ChAdOx1 nCoV-19 and showed that the transgene was the dominant transcript in E1 noncomplementing human cell lines and that unwanted transcript variants were rare [39].

We are developing a candidate vaccine with the aim to induce protective killer T cells against HIV-1 [40,41]. Through iterative improvements informed mainly by human studies of the previous HIVA [42] and HIVconsv [43] vaccines, we have arrived at bi-valent immunogens collectively designated HIVconsvX, which utilize the six most functionally conserved regions of the HIV-1 proteome assembled into unique chimeric proteins [44]. Synthetic genes coding for the HIVconsvX immunogens are delivered by a heterologous prime-boost regimen of ChAdOx1 and poxvirus MVA. Hence, two computed, mutually complementing mosaic [45] immunogens, tHIVconsv1 and HIVconsv62, are expressed by the ChAdOx1 vector as a prime. This work is about the candidate HIV-1 vaccines. We set out to assess the transcriptomic repertoire associated with the ChAdOx1-carried transgenes to increase confidence in the vaccines’ safety. Because algorithms predicting splicing are notoriously unreliable, here we analyzed the transcriptomes of the ChAdOx1.tHIVconsv1 and ChAdOx1.HIVconsv62 vaccines in infected nonpermissive and permissive human cell lines using long-read direct RNA sequencing.

## 2. Materials and Methods

### 2.1. A Brief Description of the Vector and Vaccine Inserts

The ChAdOx1 vector is derived from simian adenovirus 23 (Y25) and has deletions of the E1 and E3 early regions. In addition, E4 region open reading frames (ORFs) 4, 6 and 6/7 were replaced by the corresponding regions of HAdV-5 to improve virus yield. The vaccine inserts were designed as follows. First, curated HIV-1 proteomes of full-size proteins within group M (major) of global isolates present in the Los Alamos National Laboratory HIV Sequence Database in September 2013 were aligned and computed into two mutually complementing amino acid sequences using the Mosaic Algorithm [45]. Then, the six most-conserved regions in the HIV-1 Gag and Pol proteins were selected and assembled into tHIVconsv1 (mosaic 1) and HIVconsv62 (mosaic 2) proteins (Figure 1). These differ in approximately 10% of amino acids and together reach a perfect match of 80% of potential 9-mer T-cell epitopes (PTEs) [46,47] between the bi-valent vaccine and all the input HIV-1 isolates. This is an important feature of HIV-1 vaccines, which must effectively tackle HIV-1 variability. A small “t” in the tHIVconsv1 name indicates the presence of the human tissue plasminogen activator signal sequence [48]. Both ORFs use humanized codons [49,50] and, with adjustments, code for 895 (tHIVconsv1) and 884 (HIVconsv62) amino acids. These are inserted into the E1 locus of the adenovirus genome and are under the control of the strong early cytomegalovirus promoter [51] without the addition of upstream intron A (which was used in ChAdOx1 nCoV-19). The codon optimization for expression in human cells and reshuffle of the six conserved regions into unique chimeric proteins provide ample possibilities for newly generated features [52] including splicing signals. The ChAdOx1.tHIVconsv1 (C1) vaccine was safe and immunogenic in recently completed clinical trials (NCT04586673 and NCT04553016). ChAdOx1.HIVconsv62 (C62) entered the first-in-human trial in 2022 (NCT05604209) and there are several further trials in the clinical development program of HIVconsvX vaccines. Nevertheless, it was prudent to interrogate the C1 and C62 transcriptome in human cells to reinforce confidence in the vaccine’s safety.

### 2.2. Preparation of the ChAdOx1.tHIVconsv1 and ChAdOx1.HIVconsv62 Vaccine Stocks

The rescue and preparation of the adenovirus stocks of vaccines ChAdOx1.tHIVconsv1 (C1) and ChAdOx1.HIVconsv62 (C62) were described before [44,53]. Briefly, the C1 and C62 vaccines were grown in HEK293-derived T-REX cells (ThermoFisher, Horsham, UK), purified in a multistep process involving equilibrium ultracentrifugation through a CsCl density gradient or by anion exchange chromatography on Source 15Q resin, respectively, the presence of the transgene was confirmed by indicative PCRs, and the virus stocks were titred and stored at −80 °C until use.

### 2.3. Cell Lines and Virus Infection

The adenovirus transcriptome was analyzed in human MRC-5 cells [56], which is a genetically normal male human lung fibroblast-like cell line isolated from a 14-week-old embryo, A549 cells [57], which are a human male lung epithelial-like continuous cell line derived from carcinomatous tissue, and HEK293 cells [13], which are a human embryonic kidney epithelial cell line immortalized by the integrated HAdV-5 E1 region. All cell lines were obtained from the European Collection of Authenticated Cell Cultures (ECACC). The cells were cultured in the DMEM-10 medium (DMEM supplemented with 10% foetal bovine serum (FBS), 100 U/mL penicillin and 100 µg/mL streptomycin), infected at the confluence with each of the ChAdOx1-vectored viruses at a multiplicity of infection (MOI) of 10 to achieve synchronous infection and harvested for the extraction of RNA at 48 h post-infection for the MRC-5 and A549 cell lines and 24 h post-infection from the HEK293 cell line.

### 2.4. RNA Preparation and Sequencing

Total cellular RNA was extracted from harvested infected cells using the TRIzol^TM^ reagent (Cat. No. 15596026, Ambion, ThermoFisher, Horsham, UK) at 1 mL per 10^7^ cells, washed twice with 70% ice-cold ethanol and stored at –80 °C under 70% ethanol until use. On the day of sequencing, RNA was resuspended in sterile water, enriched for polyadenylated RNA and sequenced as described previously using the SQK-RNA002 kit and MINI06D R9 version of the flow cells (Oxford Nanopore Technologies, Oxford, UK) according to the vendors’ protocols [32].

### 2.5. Viral Transcriptome and Data Analysis

Open-reading-frame-centric data analysis of the transcriptome was described before [32]. Briefly, minimap2 [58] was used to map the transcripts to the adenoviral genome. The data were used to link the transcripts to the commonly used adenovirus transcription starts and termination locations together with the splice acceptor and donor sites. Based on these parameters, each transcript was assigned to a transcript group and its frequency was registered. These data were then used to assess the transcriptomic repertoire based on the genomic sequences, transcription start sites and features predicted to be present in the vaccine genomes and analyzed for each transcription group. The structure of each transcript group and its dominant transcript type coding for each ORF were detailed in the table of features provided. Finally, this workflow also produces a list of ORFs of proteins “not known” that are 5′ proximal for any given transcript group. Raw sequence files in fastq format, genome sequences and feature files are available at Zenodo (https://doi.org/10.5281/zenodo.36102480).

## 3. Results

### 3.1. The HIVconsvX mRNAs Dominate Vaccine Transcriptomes in Nonpermissive Cell Lines

Human A549 and MRC-5 cell lines are nonpermissive to the E1-deficient ChAdOx1 vector replication and were used to interrogate the vaccine-derived transcriptome. Cells were infected for 48 h and their total polyA RNA was enriched and sequenced using the nanopore technology. The overall sequence reads data are summarized in Table 1. In C1 vaccine-infected MRC-5 cells, the tHIVconsv1 transgene transcripts were dominant and there were no other detectable adenovirus-derived transcripts (Figure 2 and Table 2). In A549 infected cells, we were able to detect some evidence of vector backbone-derived transcripts, but these were relatively low in number (Figure 2 and Table 2). For the C62 virus-infected MRC-5 cells, again, we did not observe any full-length transcripts from the vector backbone (Figure 3 and Table 3). Similarly, in C62-infected A549 cells, we detected a range of low-level expressions of adenovirus-derived full-length transcripts. These were representative of a wide range of transcripts comparable to the C1 virus infection, but, again, they were all at low to very low levels (Figure 3 and Table 3). Low levels of rare mRNAs were detected with aberrant splicing or alternative polyadenylation site usage, but these were much below the levels of the dominant tHIVconsv1 or HIVconsv62 mRNA (Table 2 and Table 3). In contrast, there was essentially no adenovirus transcript in the MRC-5 cells (Figure 2 and Figure 3 and Table 2 and Table 3). Importantly, there was no evidence of an intron in either of the conserved mosaic transgenes. We can also confirm no detection of transcripts matching the HAdV-5 E1 transcription factors, thus confirming the absence of detectable replication-competent adenovirus (RCA) in the stock preparation.

### 3.2. HIVconsvX Vaccines Produce a Full Range of Adenovirus Transcripts in Permissive HEK293 Cells

In the HEK293 cells supplying the HAdV-5 E1 function in trans to the ChAdOx1 E1-deficient vector, a full range of adenovirus transcripts were detected (Figure 2 and Figure 3). Our ORF-centric pipeline confirmed a wide range of adenovirus transcripts and showed that the viral backbone transcripts dominate in this cell line (Table 4). Moreover, as expected, we were also able to detect adenovirus transcripts from the E1 region of the HAdV-5 genome that is integrated onto the HEK293 genome.

### 3.3. Minor Transcripts

There is a wide range of minor transcripts with unusual promoters, splicing pat-terns and/or polyadenylation sites derived from both C1 and C62 vaccines grown in HEK293 cells (Appendix A). These are usually very rare in the A549 datasets (i.e., fewer than 10 transcripts per example) (Appendix A) and typically not found at all in the MRC-5 infected cells (Appendix A). We assessed the per cent of transcripts coming from the transgene promoter with unintended splice sites to be between 2.4% and 3.4% of the total transcripts from that promoter with no splicing (Appendix A). We did observe some unusual transcripts in HEK293 cells, which included a novel promoter driving the expression of pIX from a spliced transcript (Figure 4a,b). In addition, we noted a substantial number of transcripts driven from another promoter further upstream (Figure 4c,d), which could lead to the expression of a protein with no known homology (MLGMRWALWLLRRKEPAGARGGIDPSRYLDPAFLYKVVIDSTDRDRTS). This transcript is only present in the HEK293-derived dataset. A retrospective analysis of our published data from ChAdOx1 nCoV-19 transcripts in HEK293 cells [39] revealed the presence of this transcript in that dataset too. Without matching proteomics data, it is uncertain whether this protein is made in HEK293 cells or if it plays any role during viral infection, but there are no known homologues and this transcript was not present in the HAdV-5-infected HEK293 transcriptome [32].

## 4. Discussion

In the present work, we examined RNA transcriptomes of two candidate HIV-1 vaccines, ChAdOx1.tHIVconsv1 (C1) and ChAdOx1.HIVconsv62 (C62), using direct RNA sequencing to preempt any possible safety concerns about inadvertent splicing involving the engineered synthetic transgene ORFs, which could lead to the expression of unusual protein products and cause AEs [37,38,59,60]. Our first and most important conclusion is that unintended transcripts from the transgene region are very rare and therefore unlikely to be biologically significant. For both vaccines in nonpermissive MRC-5 cells, transcription was dominated by the correct desired tHIVconsv1 and HIVconsv62 mRNAs driven by a strong eukaryotic promoter of the inserted transgene expression cassette, while there was almost a complete absence of transcription from the viral backbone. We expect that this is the most likely scenario for the C1 and C62 vaccines injected intramuscularly into human volunteers. In A549 cells, there was some low-level transcription across the vector backbone with most vector-encoded genes represented by a few but less than 5% of total transcripts. Why the A549 cell line is more permissive for vector genome transcription is unclear but may have to do with a deletion in the CDKN2A locus and loss of cell cycle control [57]. However, a direct association between this deletion and adenovirus transcription in the absence of the E1 function remains to be formally proven. In contrast, a full range of well-described and abundant viral transcripts was readily detected in the E1-transcomplementing HEK293 cell line [61]. In this cell line, transcripts with novel irregular splice patterns and no known function were also found.

The presence of these very low-level transcripts should be seen in the context of the recent advances in RNA sequencing technologies. The short-read deep sequencing of fragmented RNA (RNAseq) has generated an accurate high-resolution image of the adenovirus transcriptome [27,31]; however, quantifying individual multiply spliced RNA molecules is problematic. In contrast, direct Nanopore sequencing (dRNAseq) pulls individual RNA molecules from their polyA tail to their 5′ end and reads the sequence as the nucleotides cross the nanopore [62,63,64,65]. Importantly, both technologies matched very well on the joint splice donor and acceptor sites and cross-validated each other [32]. dRNAseq not only revealed a much greater complexity of splicing in adenoviruses [31] but also found aberrant splicing even in the human cell and herpes simplex virus transcriptomes, as well as discontinuous transcription in coronaviruses [62,63,64,65]. Our previous analysis of the wild-type HAdV-5 transcriptome showed a broad range of extremely rare transcripts made during productive viral replication amounting to up to 11,000 unique species [32]. Nevertheless, despite the vast array of low-level transcriptional “junk”, the Mastadenovirus genus of viruses remains relatively innocuous from a pathological or public health standpoint [66].

Further reassurance of the safety of the C1 and C62 vaccines comes from the fact that the transcriptomes of these vaccines were very similar to ChAdOx1 nCOV-19 [36]. During the SARS-CoV-2 pandemic, we undertook a similar analysis of the ChAdOx1 nCOV-19 vaccine’s transcriptomic repertoire [39]. Concurrent with the results reported here, SARS-CoV-2 spike mRNA by far dominated any adenovirus transcripts, which was particularly obvious in nonpermissive cell lines. Aberrant splice patterns and polyadenylation site usage were very low and judged as biologically insignificant [39]. However, comparisons between the HIV-1 vaccines in this work and the COVID-19 vaccine are not straightforward as ChAdOx1 nCOV-19 was designed with intron A upstream of the spike ORF in addition to having different ORF DNA sequences. Re-analysis of the ChAdOx1 nCOV-19 data showed that only 3.6% of the transcripts derived from the transgene promoter had unexpected splicing events that would prevent the expression of the SARS-CoV-2 S protein. This is similar to the 2.4–3.4% of unexpected splicing events we observed for the C1 and C62 transcriptomes in this work. Since its first use, over 2 billion doses of the ChAdOx1 nCOV-19 vaccine have been administered, particularly in India, South America and Africa, without any reported evidence that rare aberrant splice variants of the spike transcript caused ill effects in vaccine recipients [67,68,69].

Direct sequencing of viral mRNA is above and beyond any previous or current regulatory licensing requirements for recombinant adenovirus-vectored vaccines. We believe that this approach to studying transcription from viral vaccines is a useful additional tool to increase our understanding of the transcriptomic repertoire of experimental engineered viral vectors and provide additional assurance regarding their safety profiles.

## 5. Conclusions

The ChAdOx1-vectored C1 and C62 vaccine components are administered with a heterologous MVA HIVconsvX boost for the induction of protective T cells [43]. This HIVconsvX T-cell vaccine strategy alone and in combination with other cutting-edge tools to stop or eradicate HIV-1 is tested in a comprehensive and growing developmental clinical program encompassing a series of trials in healthy uninfected individuals for prevention and people already living with HIV-1 for a cure. The results obtained in the present work enhance the safety profile of simian adenovirus-based vaccine components C1 and C62 by eliminating, or at least significantly diminishing, one potential, previously questioned pathological mechanism and increase the confidence in these vaccines should they be widely deployed as one of the tools for HIV-1 prevention and cure.

## Figures and Tables

**Figure 1 vaccines-11-01187-f001:**
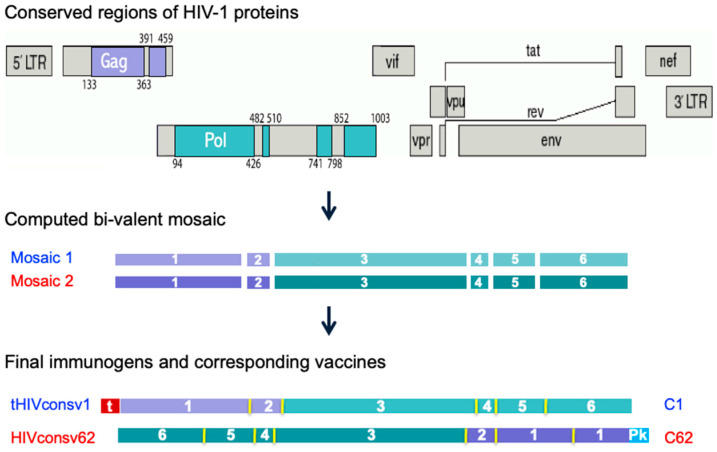
Derivation of the tHIVconsv1 and HIVconsv62 immunogens. Regions of the HIV-1 proteome selected based on their functional conservation (top) for inclusion in candidate T-cell vaccines ChAdOx1.tHIVconsv1 (C1) [44] and ChAdOx1.HIVconsv62 (C62) [53] were computed into mosaic 1 and mosaic 2 [45] (middle), which differ in about 10% of amino acids. Mosaics 1 and 2 are intended to be used together for vaccination to maximize the match of the vaccines to global group M HIV-1 isolates in terms of potential T-cell epitopes (a 9-mer amino acid window moving by 1 residue across the conserved regions). For each vaccine C1 and C62, the regions were ordered in a unique sequence to minimize induction of irrelevant junction-specific T-cell responses detected in a previous trial of a similar vaccine [54] (bottom) and the codons of the ORF nucleotides were humanized to enhance expression in vaccine recipients [49]. Regions derived from Gag are coloured purple and those from Pol are coloured green. N-terminal small “t”—human tPA signal sequence (in red); Pk—a C-terminal epitope (aka Pk or SV5 tag, in cyan) recognized by mAb SV5-P-k [55].

**Figure 2 vaccines-11-01187-f002:**
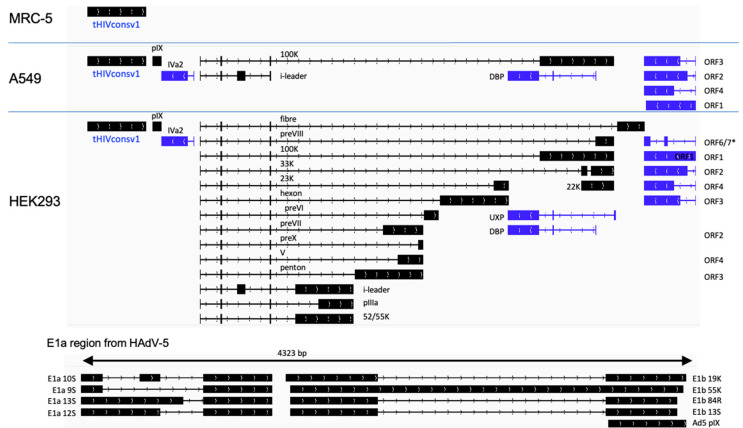
ChAdOx1.tHIVconsv1 (C1) transcriptomes in infected nonpermissive and permissive cell lines. The ChAdOx1 vector is missing adenovirus essential early region E1 and is, therefore, replication-defective in normal cells, such as MRC-5 and A549, which do not supply this function in trans. HEK293 cells carry an integrated HAdV-5 genome fragment, which supplies the E1 function and allows for the production of the full adenovirus transcriptome. A simplified schematic shows black transcripts originating from the top strand of the dsDNA viral genome (left to right) and blue RNA transcribed from the complementary strand (right to left). *, *p* < 0.05.

**Figure 3 vaccines-11-01187-f003:**
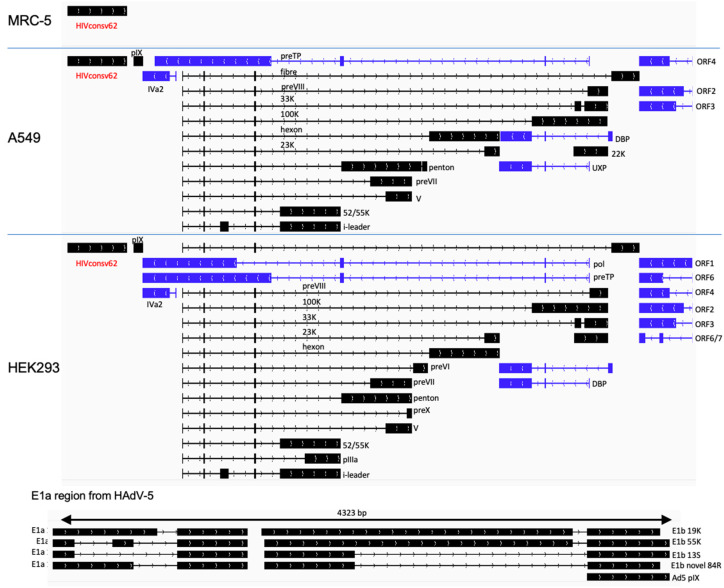
ChAdOx1.HIVconsv62 (C62) transcription maps in infected nonpermissive MRC-5 and A549 and permissive HEK293 cell lines. Please see the Figure 2 legend for details.

**Figure 4 vaccines-11-01187-f004:**
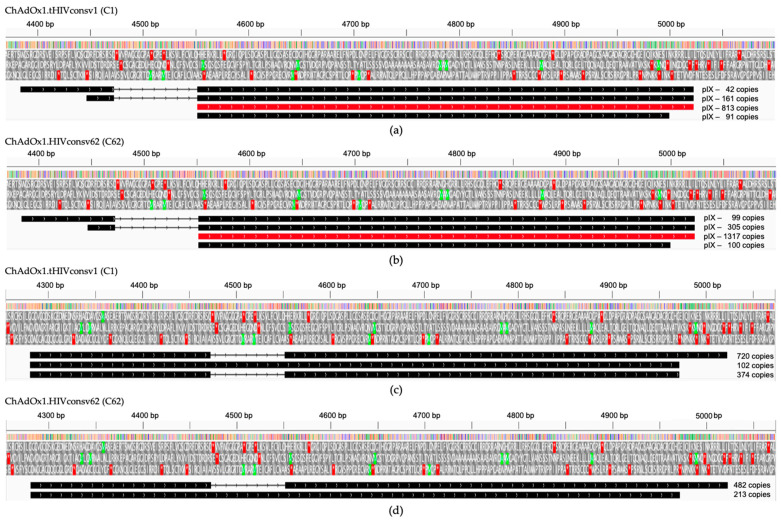
Unusual transcripts in C1- and C62-infected permissive E1-complementing HEK293 cells. A novel promoter was found to drive the expression of pIX from a spliced transcript in HEK293 cells infected with C1 (**a**) and C62 (**b**) with the main transcript highlighted in red. Several hundreds of transcripts were driven from another promoter further upstream as depicted in (**c**,**d**). Above, the schemes are the nucleotide positions in the C1 and C62 genomes. Three reading frames show the start (green) and stop (red) codon positions.

**Table 1 vaccines-11-01187-t001:** Number of reads for dRNAseq datasets.

Sample		Total Reads	Longest Read	Average Read Length	Mapped to Human Transcriptome	Mapped to Vaccine Transcriptome—Average Length of Mapped Read: Reads Mapped to E1 (HEK293 Cells Only)	Vaccine Mapped Reads as % of Human Mapped Reads
MRC-5	C1 2d *	774,687	23,376	1510	698,670	6006:2631	0.9
	C62 2d	947,095	25,754	1557	844,437	2802:2586	0.3
A549	C1 2d	1064,854	18,931	1277	961,978	8899:2542	0.9
	C62 2d	552,201	18,522	1240	482,812	7015:2402	1.5
HEK293	C1 1d	992,841	19,032	1615	487,145	343,747:2011:782	70.6
	C62 1d	1101,827	23,931	1637	505,580	460,727:2062:617	91.1

* 1d or 2d—cells were infected for either one or two days, respectively, prior to the transcriptome analyses.

**Table 2 vaccines-11-01187-t002:** Gene transcription data for the C1 vaccine in nonpermissive MRC-5 and A549 cells.

Feature	Cell Count MRC-5 Cells	% Total MRC-5 Cells	Cell Count A549 Cells	% Total A549 Cells
tHIVconsv1	3994	85.6	5698	87.66
None from the list	99	2.1	157	2.42
DBP (E2A)	0	0	52	0.80
E4 orf3	0	0	15	0.23
E4 orf2	0	0	7	0.11
pIX	0	0	3	0.05
E4 orf4	0	0	3	0.05
IVa2	0	0	1	0.02
E4 orf1	0	0	1	0.02
i-leader protein	0	0	1	0.02
100K (L4)	0	0	1	0.02

**Table 3 vaccines-11-01187-t003:** Gene transcription data for the C62 vaccine in nonpermissive MRC-5 and A549 cells.

Feature	Cell Count MRC-5	% of Total MRC-5 Cells	Cell Count A549 Cells	% of Total A549 Cells
HIVconsv62	1804	86.3	3712	80.1
None from the list	37	1.8	104	2.2
DBP (E2A)	0	0	67	1.5
i-leader protein	0	0	16	0.4
E4 orf3	0	0	8	0.2
Hexon (L3)	0	0	7	0.2
22K (L4)	0	0	6	0.1
E4 orf2	0	0	4	0.1
52/55K (L1)	0	0	4	0.1
33K (L4)	0	0	4	0.1
pIX	0	0	4	0.09
IVa2	0	0	3	0.06
Fibre (L5)	0	0	3	0.06
preVIII (L4)	0	0	2	0.04
E4 orf4	0	0	2	0.04
Penton (L2)	0	0	2	0.04
UXP	0	0	1	0.02
preVII (L2)	0	0	1	0.02
100K (L4)	0	0	1	0.02
23K protease (L3)	0	0	1	0.02
preTP (E2B)	0	0	1	0.02
pV (L2)	0	0	1	0.02

**Table 4 vaccines-11-01187-t004:** Gene transcription data for the C1 and C62 vaccines in permissive HEK293 cells.

Feature	C1 Virus		C62 Virus	
	Count	% of Total	Count	% of Total
Total of all reads	186,526		275,053	
Fibre (L5)	20,318	10.89	38,681	14.06
Hexon (L3)	18,763	10.06	42,160	15.33
preVII (L2)	17,459	9.36	27,061	9.84
tHIconsv1/HIVconsv62	14,442	7.74	6617	2.41
33K full-length (L4)	13,598	7.29	14,569	5.30
None from list	11,960	6.41	13,676	4.97
pV (L2)	9224	4.94	13,075	4.75
preX (L2)	8052	4.32	18,237	6.63
100K (L4)	7103	3.81	8259	3.00
preVI (L3)	5685	3.05	9242	3.36
i-leader protein	5082	2.72	4093	1.49
IVa2	4709	2.52	4702	1.71
DBP (E2A)	4538	2.43	3816	1.39
preVIII (L4)	4151	2.23	7838	2.85
52/55K (L1)	4073	2.18	4081	1.48
22K (L4)	3872	2.08	3636	1.32
penton base (L2)	2746	1.47	5077	1.85
preIIIa (L1)	1859	1.00	3144	1.14
pIX	911	0.49	1427	0.52
23K protease (L3)	571	0.31	959	0.35
E1b 19K	372	0.20	309	0.11
E4 orf2	285	0.15	148	0.05
E4 orf3	171	0.09	132	0.05
E4 orf6/orf7	114	0.06	90	0.03
E4 orf4	104	0.06	57	0.02
E1a 13S	78	0.04	67	0.02
E4 orf1	31	0.02	21	0.01
E1a 12S	24	0.01	24	0.01
UXP	19	0.01	315	0.11
pIX ad5	11	0.01	10	0.004
E1b 22S (E1b 55K)	3	0.002	8	0.003
E1a 10S (171R)	3	0.002	3	0.001
E1a 9S	2	0.001	2	0.0007
E1b 13S (E1b-84R)	1	0.0005	1	0.0003
E1b novel-84R	1	0.0005	1	0.0003
preTP (E2B)	0	0	13	0.005
Pol (E2B)	0	0	9	0.003
E4 orf6	0	0	4	0.001

## Data Availability

All data are provided with the manuscript.

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
