# Peer review of "Adenovirus Transcriptome in Human Cells Infected with ChAdOx1-Vectored Candidate HIV-1 Vaccine Is Dominated by High Levels of Correctly Spliced HIVconsv1&62 Transgene RNA"

_vaccines, 2023, doi:10.3390/vaccines11071187_

Round 1

Reviewer 1 Report

This is a well conducted study of the RNA transcripts generated by the ChadOx1 vector containing HIV-1 transgene inserts in different cell lines, and description of adenoviral vector RNA transcripts generated. This work is relevant to development of the ChadOx1 vector as an HIV vaccine candidate.

No major revisions recommended.

Minor revisions: Line 132: open reading instead of reding

Line 233: pre-empt instead of pre empty

Author Response

We thank the reviewer for his/her positive comments and careful reading of the manuscript. Suggested corrections were implemented.

Reviewer 2 Report

The authors present a study looking into aberrant transcripts arising from adenovirus delivery of punitive HIV immunogens. They postulate that if levels of aberrant splicing are seen for their HIV transcripts than it could inform on the possibility that a similar phenomenon was seen for the SARS-Covid vaccines. Conceptually I have a problem with their conclusions in which that state that ”These are usually very rare in the A549 dataset (i.e., fewer than 10 transcripts per example)” and “are very rare and therefore biologically insignificant”. They then extend this to say that it is unlikely to be the source of “Aberrant adenovirus RNA splicing involving transgene(s) coding for the SARS-CoV-2 spike was suggested as an etiology of rare adverse events”, however in both cases we are looking at rare adverse events. I would draw the reverse conclusion of the authors in that alterative splicing could indeed lead to some of the rare adverse events seen from the SARS vaccine. The fact that they did see some alterative splicing in the stable HIV vaccine indicates to me that this a possibility for SARS.

Perhaps what might strengthen the paper would be to get the statistics of how many rare adverse events are seen for SARS vaccine compared to their data with HIV vaccines. If they number do align, this indicates that these rare splicing events are biologically relevant in a small subgroup of patients when over a billion SARS vaccines have been given.

In its present state, I can not recommend for publications until this is clarified.

Minor comments:

1-The paper is well written and the figures are nicely presented.

2- The literature citied is not extensive and needs to be expanded.

The paper is well written and the figures are nicely presented.

Author Response

This reviewer makes some claims which we feel are a misunderstanding of our data and the background.

We are not quite sure what the reviewer means by “punitive“ HIV immunogens.

We would like to emphasise that this manuscript is about the safety of the two HIV vaccine candidates, C1 and C62, and not about the ChAdOx1 nCOV-19 vaccine (text added on line 81) and hence we did not postulate “that if levels of aberrant splicing are seen for their [our] HIV transcripts then it could inform on the possibility that a similar phenomenon was seen for the SARS-Covid vaccines.” We do not wish to inform on the safety of the ChAdOx1 nCOV-19 vaccine and we apologise for not making this clear enough. The transcription unit designed for ChAdOx1 nCOV-19 versus the ones for C1 and C62 are completely different (the ORFs differ and the presence of upstream intron A in ChAdOx1 nCOV-19 not present in C1 and C62) and thus even if aberrant splicing were observed for ChAdOx1 nCOV-19, it would not be readily transferable to C1 and C62, and vice versa. We wonder whether the reviewer is mistakenly conflating rare splice events with rare side effects - rare splice events would occur in everyone and from any adenovirus vector and they do happen in any higher eukaryote or their viruses. Text to this effect was added on line 266.

We fully endorse our conclusion that "the unintended transcripts from the transgene region are very rare and therefore biologically insignificant", whereby the most relevant cell line is MRC-5 and not A549 cells (which are defective in the CDKN2A locus) cited by the reviewer. Therefore, we disagree with the reviewer’s conclusion and apologize once again for not making our case sufficiently clear in the first place. We reiterate that text has been added on line 81 to explicitly reinforce the aim of the present work.

As requested, we have added the quantification of rare in the HIVconsvX samples on page 8 line 225. We have compared this to comparable events in the ChAdOx1 nCOV-19-infected cells on page 9 line 290.  

With respect, we once again disagree with the reviewer’s connection between the alignment of rare splice event numbers associated with SARS-CoV-2 S glycoprotein, tHIVconsv1 and HIVconsv62 ORFs and the indication that “these rare splicing events are biologically relevant in a small subgroup of patients when over a billion SARS vaccines have been given.” The authors are not aware of any published, peer-reviewed evidence linking rare splice events in the ChAdOx1 nCOV-19 vaccine with rare adverse events in vaccine recipients.

We have added a few additional references and improved the discussion with the intention to make the article content easily understandable.

Reviewer 3 Report

The paper deals with the theme of a potenti candidate HIV1 vaccine. The research is of interest and the resulta are well written.
The first paragraph of results should be more appropriately moved to the methods section. 
Moreover, findings of the study should be better discussed in the light of the studies already published on the subject. 
further considerations on the possible future development of the vaccine could be included

Author Response

As suggested, the C1 and C62 vaccine description was moved to methods.

Moreover, findings of the study should be better discussed in the light of the studies already published on the subject.

We have added as short paragraph with a few new references to the Discussion (line 269)

further considerations on the possible future development of the vaccine could be included

As suggested, a sentence has been added to the Discussion on line 312.

Round 2

Reviewer 2 Report

The reviewer appreciates the clarification as there was confusion as to the orignal purpose of the paper. A common theme in virus papers seems to be the desire to link everything to SARS/Covid. As a HIV based vaccine study, it stands on its own which was why there was confusion as to why there was discussion on SARS-Covid, leading to the confusion. I understand better now that they were not meaning to compare but used it as the emphasis for their study into HIV. The modifications to the text makes this more clear. One suggestion is to get rid of the first line of the abstract as you don't need to start with SARS. HIV studies are valid and of high interest, not everything needs to be linked to SARS. 

The reviewer still strongly disagrees with We fully endorse our conclusion that "the unintended transcripts from the transgene region are very rare and therefore biologically insignificant"  as there is not evidence to draw this conclusion due to lack of application of their HIV vaccines in patients or animal studies. Are the authors willing to tone this done and instead state are "unlikely to be" or "appear to be". I do not think such a claim is supported based on their data (they see 2-3% aberrant splicing) and we do not truly know the effect of such. It is not like they are seeing a 1 in 10,000 or 1-100,000 event. If the authors are willing to tone down this claim that these are not biologically significant than the reviewer is ok with publication.

English is fine. 

Author Response

We thank the reviewer for his/her comments and once again apologise for not making the initial manuscript sufficiently clear.

As suggested, we modified the abstract. We feel that the reason for looking at aberrant splicing should be justified, but it is now the second rather than the first sentence.

On line 120, we added the fact that C1 and C62 do not have the 5' intron A.

On line 254, we modified the text as suggested to say:

... unintended transcripts from the transgene region are very rare and therefore unlikely to be biologically significant.